

# Effect of method of deduplication on estimation of differential gene expression using RNA-seq

Anna V. Klepikova[1,2,*], Artem S. Kasianov[2,3,*], Mikhail S. Chesnokov[4], Natalia L. Lazarevich[4,5], Aleksey A. Penin[1,2,5] and Maria Logacheva[1,2,6]

[1] Institute for Information Transmission Problems of the Russian Academy of Sciences, Moscow, Russia
[2] A. N. Belozersky Institute of Physico-Chemical Biology, Lomonosov Moscow State University, Moscow, Russia
[3] N. I. Vavilov Institute for General Genetics, Moscow, Russia
[4] N.N. Blokhin Russian Cancer Research Center of the Ministry of Health of the Russian Federation, Moscow, Russia
[5] Department of Biology, Lomonosov Moscow State University, Moscow, Russia
[6] Extreme Biology Laboratory, Institute of Fundamental Medicine and Biology, Kazan Federal University, Kazan
[*] These authors contributed equally to this work.

Corresponding author
Maria Logacheva,
maria.log@gmail.com

## ABSTRACT

**Background**. RNA-seq is a useful tool for analysis of gene expression. However, its robustness is greatly affected by a number of artifacts. One of them is the presence of duplicated reads.

**Results**. To infer the influence of different methods of removal of duplicated reads on estimation of gene expression in cancer genomics, we analyzed paired samples of hepatocellular carcinoma (HCC) and non-tumor liver tissue. Four protocols of data analysis were applied to each sample: processing without deduplication, deduplication using a method implemented in SAMtools, and deduplication based on one or two molecular indices (MI). We also analyzed the influence of sequencing layout (single read or paired end) and read length. We found that deduplication without MI greatly affects estimated expression values; this effect is the most pronounced for highly expressed genes.

**Conclusion**. The use of unique molecular identifiers greatly improves accuracy of RNA-seq analysis, especially for highly expressed genes. We developed a set of scripts that enable handling of MI and their incorporation into RNA-seq analysis pipelines. Deduplication without MI affects results of differential gene expression analysis, producing a high proportion of false negative results. The absence of duplicate read removal is biased towards false positives. In those cases where using MI is not possible, we recommend using paired-end sequencing layout.

# INTRODUCTION

Transcriptome sequencing (RNA-seq) is a powerful tool for the analysis of gene expression which is more sensitive and has a higher dynamic diapason than microarrays and which became a method of choice for both model and non-model organisms. It is also a promising

method for clinical diagnostics (*Byron et al., 2016*). However, due to a higher sensitivity and a larger data output, it is more demanding with regard to experiment planning and data analysis, and more susceptible to artifacts than other high-throughput methods of gene expression analysis such as microarrays. One such artifact is the presence of duplicated reads. In RNA-seq analysis, the number of reads mapping to a certain transcript is assumed to reflect the number of molecules of this transcript in the sample. However, there are several reasons why this assumption is not always correct. First, most methods of library preparation include polymerase chain reaction (PCR) which is known to amplify some fragments more efficiently than others, depending on their GC content and length (*Aird et al., 2011*; *Dabney & Meyer, 2012*). Second, in Illumina technology, which is currently the most widely used for RNA-seq, there is a specific type of duplicate reads called optical duplicates. They arise from miscalling of a single cluster as two separate clusters. The newer series of HiSeq sequencers (3,000/4,000) uses patterned flowcells where clusters form at a fixed distance from each other. This apparently eliminates the problem of duplication caused by cluster miscalling but, because of a modified cluster generation protocol (exclusion amplification), there is a higher probability that one molecule of the library initiates two independent clusters (*Anon, 2016*; *Hadfield, 2016*). These factors can cause the number of reads to be biased and to not reflect accurately the initial transcript abundance. Such biases are especially critical for the highest and the lowest abundant transcripts.

Thus, there is a consensus that removal of duplicated reads (deduplication) is beneficial (*Dozmorov et al., 2015*). Methods of reads deduplication can be divided into two types. Methods of the first type are based on bioinformatical analysis only and are basically the removal of identical reads. Methods of the second type use extra experimental techniques, such as addition of random or quasi-random sequences, the so-called molecular identifiers (also called molecular indices or barcodes) (MI) (*Kivioja et al., 2011*; *Fu et al., 2011*; *Fu et al., 2014*). This allows one to distinguish between true duplicates and false duplicates—reads that correspond to different RNA molecules but are identical by chance—because the MI sequence is identical in true duplicates but different in fragments that are derived from different RNA molecules.

Currently, most studies implement methods of the first type. Here, there are two main approaches. In the first approach, all reads are compared against each other. As a result, clusters of identical (or nearly identical, if software can take into account sequencing errors) reads are identified, after which one read from each cluster is retained and others are discarded. A number of widely used programs are based on this approach, including BIGpre (*Zhang et al., 2011*), cd-hit-454 (*Niu et al., 2010*), FastUniq (*Xu et al., 2012*), Fulcrum (*Burriesci, Lehnert & Pringle, 2012*), JATAC (*Balzer et al., 2013*). These programs were developed and are used mainly for the 454 sequencing technology, where the number of reads is moderate (from thousands to millions) and it is feasible to make all-to-all read comparison. The second approach, widely used for high-output short-read technologies such as Illumina, is based on analysis of results of read mapping. First of all, reads are mapped on the reference sequence. After that, duplicate reads are found by searching reads mapped on the same positions in reference. This approach is used in some popular

tools, such as SAMtools (*Li et al., 2009*), Picard (http://broadinstitute.github.io/picard/), Biobambam (*Tischler & Leonard, 2014*), SAMBLASTER (*Faust & Hall, 2014*).

Although MI can be helpful (*Shiroguchi et al., 2012*), they are not used as widely as they should. The first reason for this is that despite a wide choice of protocols and kits for library preparation, only few of them use MI (to our knowledge, only ThruPLEX Tag-seq and NEXTFlex qRNA-seq kit are available for transcriptome libraries). The second reason is the paucity of software that can handle MI. Several solutions are available (*Gates & Ulintz, 2016*; *Girardot et al., 2016*) (http://www.bioscientific.com/Portals/0/White%20Papers/qRNA-Analysis.pdf) but they are non-universal and lack flexibility.

RNA-seq is a promising tool for personal medicine, in particular for oncology, where it can help to infer affected regulatory pathways, to predict the response to therapy, and to detect gene fusions (*Roychowdhury & Chinnaiyan, 2016*). This urges the development of more effective protocols, both experimental and computational. Here, we report testing the influence of different methods of deduplication (with and without the use of MI) on the estimation of gene expression in a patient with hepatocellular carcinoma and development of scripts for handling MI in the pipeline of RNA-seq analysis.

## METHODS

### Samples collection and RNA extraction

Tumor and adjacent non-cancerous liver tissues were collected and frozen in liquid nitrogen after surgical resection of T2N0M0 staged hepatitis-negative moderately differentiated HCC from 60 years old male patient with liver cirrhosis. Samples were collected with written informed consent, conforming to the ethical guidelines of the 1975 Declaration of Helsinki. The Ethics Committee of N.N. Blokhin Russian Cancer Research Center gave approval for the study. Clinical diagnosis was verified histologically. Total RNA was isolated from frozen tumor and liver samples in duplicates using PureLink RNA Mini Kit (Life Technologies) with on column DNase treatment (PureLink DNase Set; Life Technologies) according manufacturer protocol. RNA quality was checked using Agilent 2100 Bioanalyzer, all samples used for the analysis had RIN (RNA integrity number) >7.

### Library preparation

Non-ribosomal RNA was separated using Ribo Zero Gold magnetic kit (Illumina). Libraries were prepared from depleted RNA using NEBNext Ultra RNA directional kit (New England Biolabs), with following modification: instead of adapter provided in the kit we used NextFlex molecular index adapter (Bioo Scientific). Libraries were sequenced on HiSeq2000 instrument (Illumina) with read length 101 from each end of fragment.

### Sequencing data processing and construction of datasets

We generated four datasets: paired-end 100 (initial dataset, PE100)—experimental data, and three datasets derived from experimental data by cutting to 50 nt and/or by retention of one read in a pair—paired-end 50 (PE50), single read 100 (SE100) and single read 50 (SE50). All consequent steps of processing were identically applied for each dataset. First, 9 nt of reads were excluded from the read sequence by a custom script and stored for each

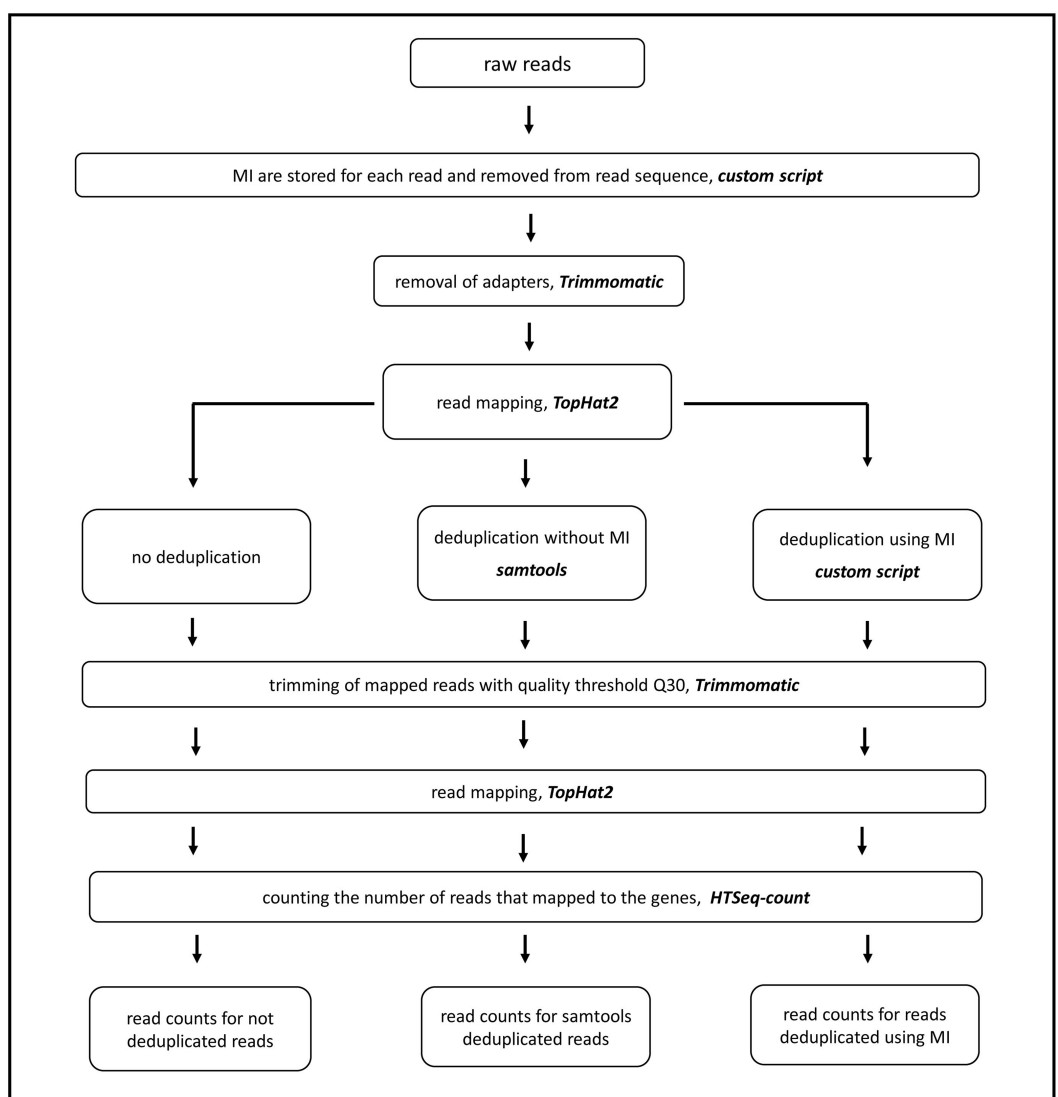

**Figure 1  Data processing workflow.**

read (respectively, for a read pair two indices were stored). Then, reads were preprocessed by Trimmomatic to remove adapter sequences (without quality control) and mapped to *H. sapiens* reference genome (processed version GRCh38) by TopHat2 (*Kim et al., 2013*). TopHat2 was executed with command line parameter:"–library-type fr-firststrand". Mapped reads were used to produce four types of read counts after following procedures: without deduplication (ND), with deduplication without using index information as realized in SAMtools module rmdup (SAM), with deduplication based on single molecular index (1MI) or two molecular indices (2MI) (Fig. 1). In the case of deduplication without using index information, duplicated reads are defined as those that have the same mapping location, irrespective of orientation (for PE reads mapping positions of both direct and reverse reads must be identical to other pairs). In the case of deduplication using index information, duplicated reads are defined as having the same mapping location and the

same MI sequence. For either dataset, a set of read counts was named as "sequencing layout_deduplication method"; for example, PE100_2MI or SE50_ND. In the case of PE100_1MI and PE50_1MI, an index from only one pair of reads was used. For SE100_2MI and SE50_2MI, both indices were counted as if two indices sequences or one longer, and 16-nt index were ligated to cDNA. ND read counts were obtained by extraction from the TopHat 2 output bam file with in-house script and then trimmed with quality threshold Q30 by Trimmomatic. Trimmed reads were mapped again with TopHat2 and read counts were generated by HTSeq. For deduplication without using index information (SAM), results of mapping were processed by the SAMtools rmdup module. Next, reads were extracted from the deduplicated SAM file with a custom script, trimmed for quality, mapped and counted as in the previous case. The 1MI and 2MI types of deduplicated read counts were the result of using a custom Perl script on the TopHat2 mapping BAM file. The script is based on two-stage algorithm. First, algorithm searches for reads with matching position on reference genome. These reads can possibly be duplicated and are referred here as duplicate-candidates. For paired-end reads, both members of pair must have positional matching in order to become a duplicate-candidate. After that, the matching of indices of duplicate-candidates is checked. If duplicate-candidates have the same index, all of them are marked as duplicate. For paired-end reads, both forward and reverse indices have to match for duplicates. Among all duplicates on certain genome position, the script extracts one representative read and discards other. After, deduplication reads were processed to generate read counts as for ND and SAM.

## DE genes identification

The samples were normalized by size factor within one set of read counts (for each type of read counts independently) (*Anders & Huber, 2010*).

DE genes were identified by the R package "DESeq2" (*Love, Huber & Anders, 2014*). The threshold for significantly differential expression was a false discovery rate (FDR) of 0.05 and fold change of 2.0.

## Real-time PCR

Reverse transcription of total RNA was conducted using random hexanucleotide primers and MMLV Reverse transcriptase (Promega, Madison, WI, USA). Quantitative Real-time PCR analysis of SCD, Glo1 and PRKAR1A genes expression was carried out using SYBR Green I PCR kit (Syntol, Moscow, Russian Federation) and iCycler Thermal Cycler with iQ5 Multicolor Real-Time PCR Detection System, data were analyzed using iQ5 Optical System Software (Bio-Rad Laboratories, Hercules, CA, USA). TATA-binding protein (TBP) was used as housekeeping reference gene. Target-specific primers were designed using NCBI Primer-Blast tool (https://www.ncbi.nlm.nih.gov/tools/primer-blast/) and synthesized by Syntol (Russian Federation). Primer sequences and PCR conditions are listed in Table S4. Real-time PCR analysis was performed on two independently isolated pairs of HCC and non-tumor RNA samples in four independent repeats for each sample. A total of 45 cycles of amplification (30 s at 95 °C, 30 s at annealing temperature (Table S4), 30 s at 72 °C) were performed and reaction specificity was checked by melt curve analysis.

Gene expression levels in each experiment were estimated using standard curve for fixed signal value. For each specimen, gene expression level was normalized to TBP expression and difference between values obtained for HCC and non-tumor samples was calculated.

## RESULTS AND DISCUSSION

For transcriptome sequencing, we have selected a HCC case for which several fresh-frozen samples of tumor and adjacent non-cancerous liver tissues were available. Samples were collected after tumor resection from the patient with an histologically verified HCC who did not receive preoperative chemo- or radiotherapy. RNA was isolated independently from two pieces of both HCC and non-tumor liver tissue. For each sample (HCC and corresponding non-tumor liver tissue, two replicates each), 33–37 millions of raw paired-end (PE) 100 nucleotide (nt) reads were obtained (for exact values, see Table S1). Each read contains a 8 nt quasi-random index sequence ($96 \times 96$ combinations), and the 9th nucleotide arises from the adenylation of cDNA fragments prior to ligation and is always A. Thus, cDNA sequence itself starts from the 10th nucleotide. Ninety-six percent of reads had indices without mismatches, and only these reads were used (Table S1). The quantities of all barcodes in the initial adaptor mix were assumed to be equal; we checked if this is remained the case after ligation and PCR. Indeed, the distribution of indices within each sample was not significantly different from uniform (Pearson's chi-squared test's $p$-value > 0.5). Also, the distribution of indices was not significantly different between samples (Pearson's chi-squared test's $p$-value > 0.2). Thus, there was no sequence-specific bias resulting in preferential ligation and/or amplification of either index.

Out of these experimental data, we generated four datasets: paired end 100 + 100 nt (PE100), paired end 50 + 50 nt (PE50), single read 100 nt (SE100) and single read 50 nt (SE50), and applied four variants of deduplication—no deduplication (ND), with deduplication without using index information (SAM) and with deduplication based on single molecular index (1MI) or two indices (2MI) (for details see methods and Fig. 1), resulting in 16 sets of read counts (Table S2).

In order to compare the effects of deduplication with and without the use of MI, we first determined the proportions of reads that remain after deduplication for SAM, 1MI and 2MI sets of read counts and compared them with ND read counts. The distributions of the proportions of remaining reads were almost identical for PE100 and PE50, as well as for SE100 and SE50 (Fig. 2 and Fig. S1). Distributions for 1MI and 2MI were identical for PE100 and PE50 and have a significantly ($t$-test $p$-value < 2.2e−16) lower means compared to that of SAM (0.6 vs. 0.62, respectively). In the case of single reads independent of read length, 1MI had a slightly lower mean than 2MI (0.59 and 0.58, $t$-test $p$-value < 2.2e−16), and distributions of fraction of remaining reads for SAM were considerably wider with a lower mean (0.48).

A crucial question is whether these reads are evenly distributed across the genes with different expression levels. True duplicates can occur for any library fragment. Thus, one can expect that if only true duplicates are removed, the distribution does not depend on the expression level. In contrast, false duplicates—reads that arise from different RNA

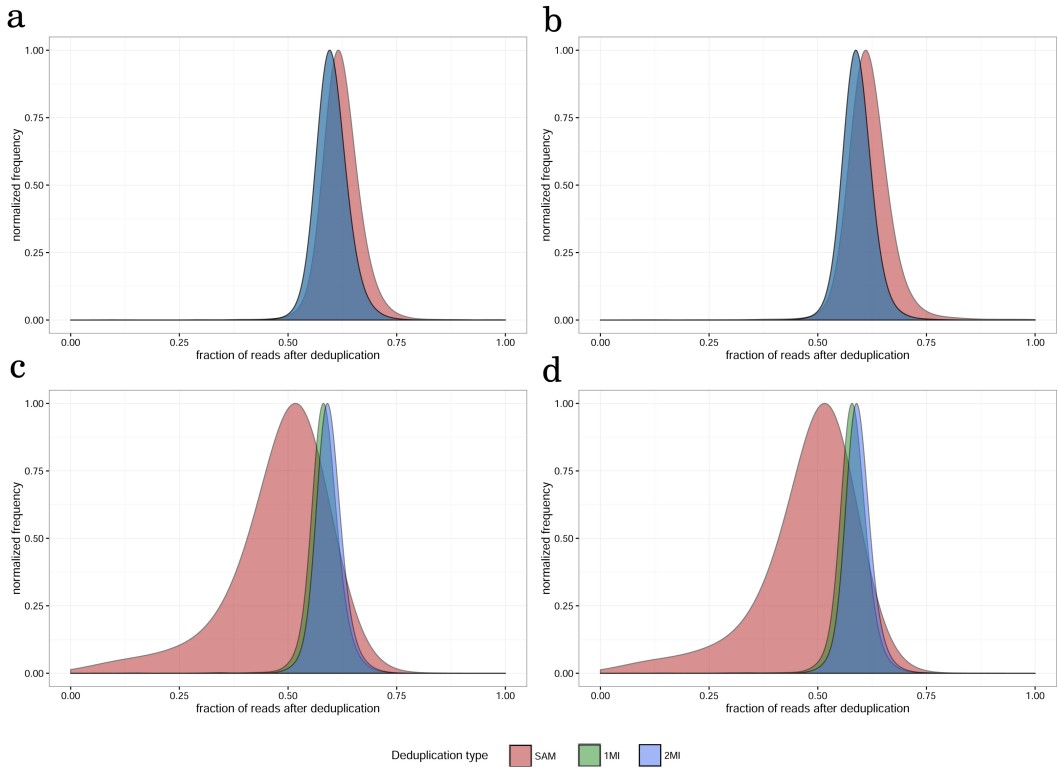

**Figure 2** **The distribution of fraction of reads remaining after different types of deduplication (read count per gene after deduplication/read count per gene after deduplication) for (A) paired-end, 100 nt, (B) paired-end, 50 nt, (C) single-end, 100 nt, (D) single-end, 50 nt read datasets for non-tumor tissue sample, replicate 1.**

molecules but are identical—are preferentially observed in highly expressed genes. In order to infer the effect of deduplication on genes with different levels of expression we first calculated Reads (for paired end datasets—fragments) Per kilobase of transcript per million mapped reads (RPKM/FPKM) values for ND dataset (RPKM was taken as a measure of gene expression because it is independent of length). Then, for each dataset we calculated the fraction of remaining reads across all RPKM.

As in the previous case, PE100 and PE50 show almost identical results (Figs. 3A, 3B and Fig. S2). For single read data, the results are also very similar for different read lengths (SE100 and SE50). For PE reads the proportion of reads that remain after deduplication for 1MI and 2MI did not depend on the level of gene expression, whereas for SAM it was approximately 1.5 times lower for the most highly expressed genes (Figs. 3A, 3B), indicating that a substantial proportion of the removed reads are not true duplicates but independent reads. In the case of single-end reads, all the three methods of deduplication produced different results. For SE100_2MI and SE50_2MI the proportion of remaining reads was identical to PE_2MI deduplication and did not depend on gene coverage (Figs. 3C, 3D). The 1MI for single-end reads produced a slight decrease of the proportion of retained reads for genes with high coverage (Figs. 3C, 3D). The SAM type of deduplication on single reads demonstrated a dramatic decrease of the proportion of remaining reads
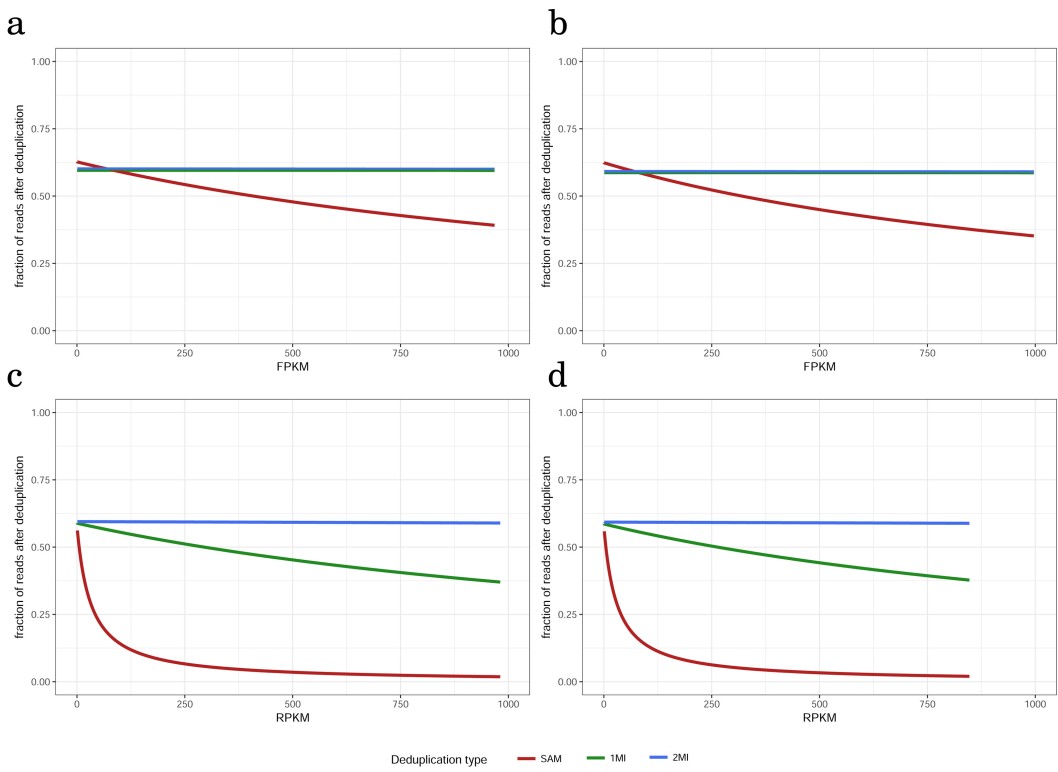

Figure 3 **The dependence of fraction of reads remaining after deduplication on gene expression level for (A) paired-end, 100 nt, (B) paired-end, 50 nt, (C) single-end, 100 nt, (D) single -end, 50 nt read datasets non-tumor tissue sample, replicate 1.**

already at RPKM 100 (Figs. 3C, 3D). This shows that most reads removed in SE50_SAM and SE100_SAM datasets are not true duplicates.

In order to assess the impact of deduplication on the analysis of gene expression, differentially expressed (DE) genes were identified between HCC and non-tumor liver samples for each set of reads using DESeq2 (*Love, Huber & Anders, 2014*). PE100_2MI set was used as a standard due to the best performance of deduplication algorithm which uses two indices on 100 nt pair-end data. In PE100_2MI set, 2,656 down- and 1,829 upregulated DE genes were discovered in HCC sample compared to non-tumor liver tissue. The number of DE genes for all but SE50_SAM and SE100_SAM sets of reads was very close to PE100_2MI standard, with the difference being below 7%. For SE50_SAM and SE100_SAM the number of DE genes was lower than for the standard, for down- and upregulated genes by 20 and 30%, respectively (the full list of DE genes for all read sets is presented in Table S3). The percentage of gene lists intersection with PE100_2MI was calculated for each read set. Two values were assessed: the percentage of DE genes only in PE100_2MI (false negative, FN) and only in the compared read set (false positive, FP). Although having similar number of DE genes, sets of reads varied in their value of intersection with the standard (Fig. 4 and Table S3). PE100_1MI was almost identical to the standard and had low FN and FP percentage, 0.1 and 0.2, respectively. The rate of FN was also low (below 5%) in PE100_SAM, PE100_ND, SE100_2MI, SE100_1MI, and
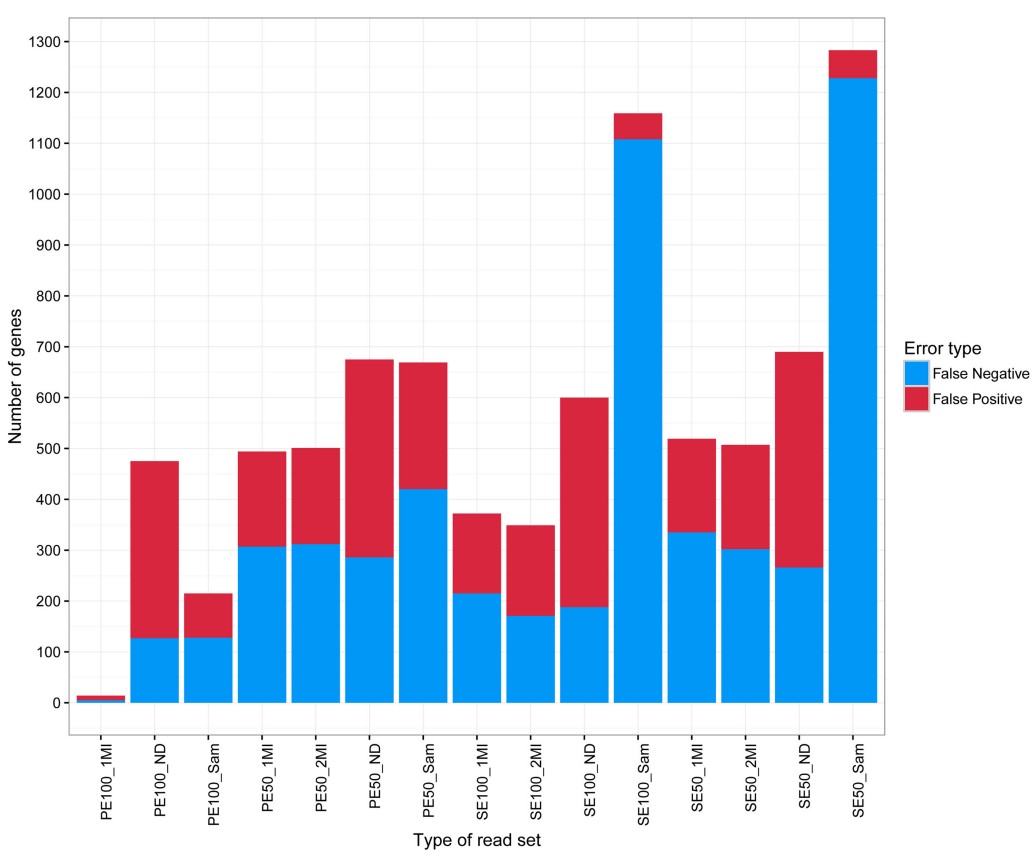

**Figure 4** **Errors in DE genes detection for different reads length and types of deduplication (PE100_2MI used as a standard).**

SE100_ ND. Consistently with a lower number of DE genes, SE50_SAM and SE100_SAM had extremely high levels of FN, 24.7% and 27.4%, respectively (Fig. 4). As expected, FNs are the most pronounced effect of deduplication. All 2MI and 1MI read sets and PE100_ SAM had a FP rate less than 5%. SE100_SAM and SE50_SAM had a lowest percentage of FP (1.1 and 1.2%, respectively). Only the absence of deduplication (Table S3 ) led to high FP rate for all types of reads (from 7.6 to 9.5%, Fig. 4). As mentioned above, for SE reads deduplication with SAMtools led to a sharp decrease of the proportion of remaining reads for high-coverage genes. Thus, we concluded that most of FNs affected highly expressed genes. Indeed, for example, gene ENSG00000000971 had 36 732,41 and 9 830,56 of mean-normalized read counts in PE100_2MI and 58 201.08 and 15 939.78 at PE100_ND for non-tumor liver and HCC tissues, respectively, and in both cases changes its expression is significant with fold change ~3.6 (Table S2). For SE50_SAM, the mean expression level of this gene is 3697.15 and 2068.26 for liver and HCC tissues, respectively, and the gene is not detected as DE. Such distortion of expression profiles of highly expressed genes is typical for the use of single reads deduplicated by SAMtools (median RPKM for FN is 8.3 in HCC and 4.7 in liver tissue, while median RPKM for genes that are identified as DE in both PE_2MI and SE_SAM is 1.0 in HCC and 1.3 in liver tissue). We chose three such genes, encoding stearoyl-CoA desaturase (*SCD),* glyoxalase I (*Glo1*) and protein kinase CAMP-dependent

type I regulatory subunit alpha (*PRKAR1A)* that are significantly upregulated in tumor samples in all sets of reads with an exception of SE100_SAM and SE50_SAM, for real-time PCR assessment. Fold changes of these genes calculated by real-time PCR data well match sequencing-based ones (Table S4). Therefore, for accurate analysis of highly expressed genes using single-end reads deduplication using MI is essential and two MI or one longer MI are preferable. For experiments with single read data without MI we do not recommend deduplication at all; however, results should be interpreted with caution because they can include false positives.

Incorrect results of estimation of differential expression can lead to erroneous biological interpretations. Among DE genes with high levels of expression that were identified in PE100_2MI and PE50_2MI sets of read counts, but not in SE100_ SAM and SE50_SAM ones, a significant number of genes which activation or repression was previously reported to be associated with malignant transformation or particularly HCC pathogenesis were observed. According to cBioPortal (*Gao et al., 2013*), analysis of TCGA provisional Liver Hepatocellular Carcinoma dataset which included 442 samples, 12 of 32 highly expressed genes with more than 3-folds activation in HCC identified by PE100_2MI but not by SE100_SAM and SE50_SAM read count sets were overexpressed or amplified in more than 10% of HCC cases. These genes encode EEF1D elongation factor deregulated in multiple cancers and modulating proliferation and epithelial-mesenchymal transition in oral squamous cell carcinoma; Glo1 glyoxalase required for HCC cells proliferation and survival which is upregulated or amplified in 48% of HCC samples; liver-specific target of WNT signaling pathway glutamine synthetase GLUL; PRKAR1A regulatory subunit of cAMP-dependent protein kinase encoded by TSE1 locus associated with regulation of hepatic differentiation, and some others (*Flores et al., 2016*; *Zhang et al., 2014*; *Zucman-Rossi et al., 2007*; *Boshart et al., 1991*).

Notably, SE100_SAM and SE50_SAM procedures failed to detect significant overexpression of multifunctional adapter protein-encoding gene *SQSTM1* which regulates cellular growth, antioxidant defense, autophagy, and survival through interplay with numerous signaling molecules and pathways such as Nrf2, mTORC, ERK1, NF-kB, ubiquitin and many others (*Taniguchi et al., 2016*). Recently, SQSTM1 was shown to accumulate in liver with nonalcoholic steatohepatitis (NASH) and hepatocellular carcinoma and promote liver cancer through Nrf2-dependent metabolic reprogramming (*Saito et al., 2016*). Among significantly downregulated genes not identified by SE100_SAM and SE50_SAM methods, there were such genes as *TMEM107*, which encodes a transmembrane protein required for normal regulation of sonic hedgehog (Shh) signaling involved in hepatocarcinogenesis; *TXNIP*, whose product is a potential tumor suppressor thioredoxin-interacting protein which participates in regulation in the course of oxidative stress, apoptosis, and cell proliferation in hepatic cells; secreted proteins transthyretin (*TTR*), orosomucoids (*OSM*)1 and 2, serum amyloids A (*SAA*) 1 and 2, serum amyloid P component (*APCS*), hemopexin (*HPX*) and complement factor H (*CFH*) whose deregulation is associated with a variety of liver pathologies and carcinogenesis or is linked to cancer prognosis (*Christopher et al., 2012*; *Hamilton et al., 2014*). In contrast to 100 nt read length variants, the usage of the 50 nt read length datasets (both PE and SE with

any variant of deduplication or without it) failed to detect downregulation of neuronal growth regulator 1 *NEGR1* recently described as a commonly regulated gene in multiple cancers but not previously reported to be deregulated in HCC. NEGR1 is raft-associated membrane protein involved in cell-to-cell recognition and adhesion that was shown to suppress cell proliferation, migration and invasiveness in ovarian adenocarcinoma cell line (*Kim et al., 2014*). A total of 50 nt read length datasets also failed to reveal deregulation of additional cancer-associated genes, particularly *CDCA3* (cell division cycle associated 3) regulating cell cycle and modulating drug sensitivity in several types of cancer including HCC; formin family *DIAPH3* influencing amoeboid migration of tumor cells and sensitivity to taxanes in breast cancer patients; candidate HCC prognostic factor *KIAA0101* involved in cell cycle progression and DNA repair regulation; UDP glucuronosyltransferase *UGT2B28*; breast carcinoma amplified sequence *BCAS4* and some others (*Jang et al., 2013*; *Morley et al., 2015*; *Abdelgawad, Radwan & Hassanein, 2016*).

## CONCLUSION

We showed that the method of deduplication greatly affects all steps of RNA-seq data processing, from removal of duplicated reads to gene expression estimation. The type of reads (single read or paired-end) is also important, in contrast to read length. For single-end data, only deduplication with molecular indices guarantees correct identification of DE genes. The usage of pair-end reads and deduplication based on two indices can markedly increase the sensitivity and accuracy of DE analysis and identification of new diagnostic and prognostic markers and therapeutic targets for NGS-based cancer detection and treatment.

## ACKNOWLEDGEMENTS

The authors are grateful to Alexey S. Kondrashov for proofreading and editing the manuscript.

### Funding
The study was supported by the Ministry of Education and Science of Russia (project # 14.607.21.0049, ID RFMEFI60714X0049). The funders had no role in study design, data collection and analysis, decision to publish, or preparation of the manuscript.

### Grant Disclosures
The following grant information was disclosed by the authors:
Ministry of Education and Science of Russia: #14.607.21.0049, ID RFMEFI60714X0049.

### Competing Interests
The authors declare there are no competing interests.

## Author Contributions

- Anna V. Klepikova analyzed the data, wrote the paper, prepared figures and/or tables.
- Artem S. Kasianov analyzed the data, prepared figures and/or tables.
- Mikhail S. Chesnokov performed the experiments.
- Natalia L. Lazarevich performed the experiments, contributed reagents/materials/analysis tools, reviewed drafts of the paper.
- Aleksey A. Penin conceived and designed the experiments, reviewed drafts of the paper, interpreted the results.
- Maria Logacheva conceived and designed the experiments, contributed reagents/materials/analysis tools, wrote the paper, interpreted the results.

## Human Ethics

The following information was supplied relating to ethical approvals (i.e., approving body and any reference numbers):

The Ethics Committee of N.N. Blokhin Russian Cancer Research Center gave approval for the study.

## DNA Deposition

The following information was supplied regarding the deposition of DNA sequences:

Raw reads are deposited in NCBI Sequence Read Archive under bioproject PRJNA354977.

## Data Availability

GitHub: https://github.com/ArtemKasianov/UMKA.

## Supplemental Information

Supplemental information for this article can be found online at http://dx.doi.org/10.7717/peerj.3091#supplemental-information.

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
