# Peer review of "Effect of method of deduplication on estimation of differential gene expression using RNA-seq"

_PeerJ, doi:10.7717/peerj.3091_

## Round 0.1 · original submission · Major Revisions

Please, provide special attention to the notes of the reviewer 2, especially provide the reader with the definition of read duplication and its principal causes. Please, discuss also the trimming of reads.

As both the reviewers have underlined the unacceptable language quality, I'm asking you to perform a careful language edition, preferably by inviting a native English speaker.

·

Basic reporting

The manuscript is written in a clear, unambiguous, professional English. The introduction provides all necessary details on the state of the art and explains well the main motivation of the study. The authors provide all necessary references. The figures are relevant, they are generally well labelled and described.

Major comments:

Raw data supply:
The authors state that “Raw reads are available in NCBI, bioproject #PRJNA354977” (Ln 291). However, I could not find this dataset. Can the authors provide a direct link?

Ln 102: It is not clear how the read count distributions were obtained. It was distribution over what set of experiments/samples? Was it for read count per gene?

Minor comments:

Ln 43: PCR => Polymerase chain reaction (PCR)
Ln 91 & 93: wasn’t => was not
Ln 100: without it we first => without it, we first
Ln 103: Fig.2 => Fig. 2
Ln 114: RPKM => Reads Per Kilobase of transcript per Million mapped reads

Ln 114: Should the authors use FPKM instead of RPKM for PE reads?

Ln 120: (Fig.3a,b) => (Fig. 3a, 3b)
Ln 124 & 125: Fig.3c,d => Fig. 3c, 3d.
Ln 126: RPKM 100 (Fig.3c,d). (Fig.3c,d). => RPKM 100 (Fig. 3c, 3d).
Ln 191: “Tmem107” => “TMEM107" as it is a human gene.
Ln 244: “Tophat2” => “TopHat2” here and throughout the text; and in Figure 1.

Figure 1: Use capital first letters in every box; Tophat2 => TopHat2
Figure 4: Please, add ticks for every 250, 200 or 100 genes to improve the readability of the figure
Suppl. Figure S1: Please provide the color code in the legend or on the figure itself.

Experimental design

The research question is well defined, the results will be useful for the research community. The experimental design corresponds to the question addressed and is fully described in the manuscript.
Ethical standards are met. The Methods section provide sufficient details.

Validity of the findings

The data are robust, the results are statistically sound. Conclusions are well stated.

Reviewer 2 ·

Basic reporting

The article is written in English as it is required but I would suggest the authors should go through the text one more time to correct errors like:

Ex1: Abstract: “In cases where the use UMI is not possible, we recommend to use paired-end sequencing layout.” => “In cases where the use of UMI is not possible, we recommend using the paired-end sequencing layout”.

Ex2: “It is more sensitive and has higher dynamic diapason than microarrays and became a method of choice for both model and non-model organism and is a prospective method for translating into clinical diagnostics (Byron et al., 2016).”
Question for clarification:
=> “It is more sensitive and has higher dynamic diapason than microarrays and became a method of choice for both model and non-model organisms and is a prospective method for translating it (?) into clinical diagnostics (Byron et al., 2016).
Suggestion:
=>“It is more sensitive and has a higher dynamic range than microarrays which is why it became the method of choice for both model and non-model organisms as well as prospective method for clinical diagnostics (Byron et al., 2016).”
OR
=>“It is more sensitive and has higher dynamic range than microarrays and became a method of choice for both model and non-model organisms. It is the prospective method for clinical diagnostics (Byron et al., 2016).”

Ex3: “However due to library that include PCR in most (if not all) currently used methods, the ratio of initial transcript abundance and read number may be biased” - this sentence is hard to read...
Suggestion: “However, since in most (if not all) currently used methods the libraries include PCR, the ratio of the initial transcript abundance and the number of reads might be biased.”

Ex4: “Though the use of UMI is favorable (Shiroguchi et al., 2012), they are not as widely use as they deserve”
Ex5: “ …with read leangth 101 from each end of fragment.”
Ex6: Fig1 - please revise texts in all boxes
You may want to consider "count the number of reads that mapped to a gene", mapped read count" and other variants, but in any case "mapped on gene" must be replaced with "mapped to a/the gene"

etc, etc, etc …

Also wording in a number of places should be improved to make the statements more clear:
ex: “Deduplication without UMI alters results of differential gene expression analysis, creating high fraction of false negative results.” – “alters” or “affects”?
Suggestion:
“Deduplication without UMI affects the results of the differential gene expression analysis, creating a high number of false negative results”

These are just a few examples, but I think that overall the text is not always clear and can sometimes come through as ambiguous. In a number of spots articles and transition words need to be added in to improve the flow.

The "Results and discussion" part starts with the sentence:
For each sample (HCC and corresponding non-tumor liver tissue, two replicates each) 33-37 millions of raw 100+100 reads were obtained (for exact values see Table S1).
I suggest to add few more words describing samples and how they were chosen

Experimental design

1. A whole number of abbreviations is used in the text before they are defined (PE, SE etc). Most of them are intuitively clear but it is better to introduce them before they are widely used in the text.

2. It is not described what causes read duplication (duplicated reads are not always from PCR amplification artefacts) and how duplication is defined:
- Reads that are exactly (almost?) the same and on the same strand? and get aligned at the same place? – all of the above? None of the above? etc
Especially in case of PE reads (both reads – direct and reverse - must be identical to other pairs or just one is sufficient? Means, alignments that start at the same locations for both read 1 and read 2 or just one read?).
Duplicated read is not the same as duplicated transcript. This should also be clearly stated.

3. Question regarding the Data processing workflow: why not to run quality trimming before read mapping?

Validity of the findings

no comment

Additional comments

The article is dedicated to a very important subject and good robust tools for read deduplication are in great demand, but I have to suggest critical revision of both English language as well as the structure of the article before it is published

---

## Round 0.2 · accepted · Accept

Please take into account the notes of the reviewer. I'm also would like to remind you that we are waiting for the #PRJNA354977 data (reads etc) to be open, as it was stated as 'subject of acceptance'

·

Basic reporting

Minor comments:

Ln 194 & Figure 2:

“Ln 102: It is not clear how the read count distributions were obtained. It was distribution over what set of experiments/samples? Was it for read count per gene?
This is the distribution of ratio (read count per gene after deduplication/read count per gene after deduplication). Read count were determined using HTSeq”.

This information should be added to the manuscript (text and Figure 2).

Ln 67. As a result,clusters => As a result, clusters

Experimental design

The article meets standards.

Validity of the findings

The article meets standards.